# Artificial Intelligence Systems Assisting in the Assessment of the Course and Retention of Orthodontic Treatment

**DOI:** 10.3390/healthcare11050683

**Published:** 2023-02-25

**Authors:** Martin Strunga, Renáta Urban, Jana Surovková, Andrej Thurzo

**Affiliations:** Department of Orthodontics, Regenerative and Forensic Dentistry, Faculty of Medicine, Comenius University in Bratislava, 81250 Bratislava, Slovakia

**Keywords:** orthodontics, AI, ChatGPT, AI Treatment Assessment, Teledentistry Cephalometrics

## Abstract

This scoping review examines the contemporary applications of advanced artificial intelligence (AI) software in orthodontics, focusing on its potential to improve daily working protocols, but also highlighting its limitations. The aim of the review was to evaluate the accuracy and efficiency of current AI-based systems compared to conventional methods in diagnosing, assessing the progress of patients’ treatment and follow-up stability. The researchers used various online databases and identified diagnostic software and dental monitoring software as the most studied software in contemporary orthodontics. The former can accurately identify anatomical landmarks used for cephalometric analysis, while the latter enables orthodontists to thoroughly monitor each patient, determine specific desired outcomes, track progress, and warn of potential changes in pre-existing pathology. However, there is limited evidence to assess the stability of treatment outcomes and relapse detection. The study concludes that AI is an effective tool for managing orthodontic treatment from diagnosis to retention, benefiting both patients and clinicians. Patients find the software easy to use and feel better cared for, while clinicians can make diagnoses more easily and assess compliance and damage to braces or aligners more quickly and frequently.

## 1. Introduction

Assistive technologies and automated systems are high-tech elements that are every day reshaping workflows of modern healthcare. Assistive technologies, including virtual reality, are designed to improve or maintain a person’s functioning so that they can participate in all aspects of life [1,2,3]. Automated systems empowered with Artificial Intelligence (AI) can support healthcare decision-making, therapy, and rehabilitation and can also help prevent treatment errors. These technologies can be used individually or can be interconnected to create assisted living solutions or enable rehabilitation at home [4]. Artificial intelligence is essential for advanced computer aided diagnostics’ [5] appropriate integration of social robots with the potential to bring benefits to aged care [6] and also future hybrid exoskeleton systems [7]. Various telemonitoring systems will benefit from the AI evaluation of sensor data from mobile phones or wearables, e.g., patient movements in the early diagnosis of Parkinson’s disease [8]. AI is not only the future of advanced robotics in healthcare [9,10], but it is also cornerstone of advanced digital radiology [11] in dentistry, including the orthodontic specialty [12,13,14]. The editorial of the International Journal of Environmental Research and Public Health from September 2022 by Giansanti summarized what is expected from an AI-based system [2] in the public health domain.

Today’s rapidly growing desire by dental practices to increase the effectiveness of their treatments has led to the development of numerous tools to achieve this, such as Dental Monitoring software (DM) (Dental Monitoring Co., Paris, France), StrojCHECK by Sangre Azul (3Dent medical Ltd., Bratislava, Slovakia), White teeth, etc. DM is a combination of artificial intelligence and telemedicine that enables easy daily collaboration and communication between the dental practice and the patient via a smartphone software app. This facilitates the coordination and execution of each step and the monitoring of the achieved goals throughout the treatment. It is feasible for both parties to use the maximum potential of this tool. There is an increasing demand for health apps not only in orthodontic dentistry but also in other medical specialties [15,16].

The possibilities of health apps are immense, ranging from promoting an active, healthy lifestyle, assisting with nutrition, preventing diabetes and high blood pressure, and treating depression to changing behavior to stop smoking and drinking alcohol, taking medications regularly, etc. They also enable monitoring and adjustment of calorie intake and output. For this purpose, additional devices such as wristbands and smartwatches are often used alongside smartphones. These sensor systems vary from accelerometers, barometers, geosensors, heart rate sensors, etc. [17,18]. Additionally, current studies have shown that telemedicine also provides a way to improve primary care accessibility, as it can decrease the time to specialty consultation, reduce the number of patients on the waitlist, and it allows the more urgent cases to reach a specialist sooner [19]. The high technological level of sensors in smartphones have led not only to dental monitoring but also to utilizations of optical scanning for 3D face morphology registration [20,21].

This application of telemedicine, specifically teledentistry, has proven to be increasingly popular and acceptable amongst not only adolescent and child patients but also in adults [22]. For some clinical applications of advanced 3D-printed appliances in children with craniofacial syndrome, regular home telemonitoring would be extremely valuable and would minimize the potential risks of appliance damage and treatment failure in complicated cases, such as Pierre Robin Sequence patients with 3D-printed palatal plates or common orthodontic patients with 3D-printed distalizers [21,23,24]. This also brings an economic and efficiency aspect to the usage of various types of telehealth software. Current data from after the COVID-19 pandemic show that treatments monitored with a DM app required 33.1% less appointments than patients without monitoring. In addition, the duration of the first phase of treatment was reduced by 1.7 months on average for the DM group and, finally, although without clinically significant relevance (less than 0.5 mm or less than 2°), there was an increased accuracy of movements expressed on maxillary and mandibular anterior teeth when compared to predicted positions [25,26].

Studies have shown that their use is perceived as feasible for several reasons: the first, and particularly important, reason is the behavioral impact on the patient during usage of these tools. It has been proven that a patient’s engagement in the treatment is considerably improved as a direct effect of working with the app. As a result, better compliance is expected; hence, the outlined outcome should be improved accordingly. Compliance is, apart from the quality of treatment planning and difficulty of the teeth movements, increasingly one of the most crucial aspects of achieving treatment goals, especially for aligner treatments, which are on a significant popularity rise. Furthermore, when a patient is being self-scanned on a 4-, 7-, 10-, or 14-day basis, he is also aware that the hygienic status of his teeth will be assessed and visible to the doctor, assistants, and even third party (the software staff as well), which, overall, leads to improvement in his dental hygiene [27,28,29,30].

The software uses a knowledge-based algorithm that evaluates the data patients send to the app after taking a series of photos with their smartphone. An automatic preset for feedback and comments is then sent back to the patient, containing a lot of data for the patient about their current dental status [31].

Unlike other telecommunications systems such as Skype, Google Duo, Zoom, and others [14], which cannot provide a standardized evaluation of the clinical situation, the DM system provides process automation through a knowledge-based algorithm that is based on a combination of robotic and deep learning processes, with information systems that act like a semi-intelligent user [25,32].

The aim of this article was to investigate the use of advanced AI software in orthodontics, particularly for the purposes of CBCT diagnosis and assessment, treatment progress assessment, and outcome stability in the follow-up phase. We evaluate the accuracy and efficiency of these AI tools compared to conventional methods and discuss the potential benefits of using such software in orthodontic practice, including the ability to closely monitor each patient, set specific treatment goals and track their achievement, and detect changes in occlusion, jaw translation, and tooth movement.

The secondary objective was to summarize reported limitations of implemented AI-powered systems in orthodontics.

## 2. Materials and Methods

### 2.1. The Research Question

The question for the literature research was defined specifically enough to allow the review team to identify relevant studies, but broadly enough to capture the full scope of the topic being reviewed.

How are AI systems currently assisting the assessment of the treatment or retention of orthodontic treatment clinically implemented, and what are their advantages and limitations?

### 2.2. The Search Strategy

The search strategy aimed to identify all relevant studies on the topic being reviewed. This involved searching databases and grey literature to ensure that this review was as comprehensive as possible. For this review, PubMed, Scopus, the Web of Science—Core Collection, and Google Scholar were queried.

The query was developed in dialogue with AI ChatGPT 3.5 Dec 15 Version (OpenAI Inc., San Francisco, CA, USA). Databases were queried on 20 December 2022 with the following query:


*(orthodontic treatment OR orthodontics) AND (artificial intelligence OR machine learning OR deep learning) AND (assessment OR evaluation OR prediction) AND (course OR retention OR outcomes)*


The definition of the query was suggested upon drafts of this review title, abstract, and defined research question and was accepted by all four evaluators. This search query would find articles that discuss the use of artificial intelligence systems in evaluating the course and retention of orthodontic treatment and contain the relevant terms “orthodontic treatment”, “artificial intelligence”, “evaluation”, and “course” or “retention”.

### 2.3. The Review Process

All studies returned by search were analyzed for duplicities followed by analysis from four evaluators for title and abstract evaluation. Only studies relevant to the topic were selected, and relevant data were extracted.

## 3. Results

All articles below dating before 2020 were eliminated from the study, as only the most contemporary and relevant data were to be gathered.

We excluded 17 articles that complied with queried keywords but were not addressing the topic even marginally. Table 1 shows most cited articles relevant to the queried keywords.

### 3.1. Cephalometric Landmark Detection and Placement by Artificial Intelligence

Multiple studies confirmed a wide range of software enabling recognition and detection and automatic placement of cephalometric landmarks, detecting pathologies using CBCT images, pathologies ranging from tumors, cysts, periapical lesions, caries, supernumerary teeth, tissue alterations as present in infectious processes, and abscess formations. In various measurements, they compared the accuracy of these evaluations to the skills of a trained dentist, all showing more than 95% compatibility with the findings of the dentists [33,35,36,37,40,49,50].

Juerchott et al. are also studying whether MRI can serve as an alternative to CBCT for 3D cephalometric analysis. Mean values were found to be equivalent, which supports this thesis, which could possibly reduce radiation exposure for many patients [44].

Moreover, segmentation of the facial skeleton was carried out by automatized MS-D convolution networks then compared to a segmentation set by orthodontists; the mean difference was insignificant, whereas the amount of time needed for segmentation was about 5 h for 1 CBCT for an orthodontist and 25 s for the CNN. This study showed that an incredible amount of time was possibly saved by this AI [38].

Ren et al. gathered data that also claim AI and deep machine learning is not already utilized for a cephalometric landmark, but it is already being used for determination of cervical vertebrae stages, oral cancer detection, cancer margin assessment, its prognosis, dental caries detection, root morphology, the presence of periapical lesions, and facial attractiveness evaluation [46].

### 3.2. Dental Monitoring System Applications

A study by Dallesandri et al. studied the approach of patients and dentists toward a DM system throughout their orthodontic treatments. Collected data showed that all dentists judged telemonitoring positively, as 96.25% of them considered telemonitoring indicative of high-tech and high-quality treatment, and 100% considered it a way to reduce the number of in-office visits. In addition, 97.5% of patients judged telemonitoring positively; 81.25% of them considered telemonitoring indicative of high-tech treatment; 81.25% declared themselves to be interested in reducing the number of in-office visits through telemonitoring. Telemonitoring was assessed as plausible both by patients and dentists; it was also understood as a high-tech tool that could improve quality and effectiveness of the treatments. Both groups were also pleased by possibly reducing the number of in-office visits. However, additional funding for this utility from the side of the patient was less welcomed, and compliance would be put to the question if such was the case [47].

Caruso et al. carried out a two-case study where they assessed treatments of patients using DM. Both patients displayed good compliance and successfully reached all established treatment goals. The needed movements were difficult to achieve, yet, owing to being able to be monitored, they completed treatment quickly; they both followed a seven-day exchange protocol, which is slightly faster than the usually observed treatment speed. There were phases in treatment when it was necessary to prolong the time on each aligner, while maintaining adequate tracking. After this period, the speed adaptively returned to the previous schedule. Patients assessed that monitoring was easy to use; it detected debonding auxiliaries and thus improved quality of the treatment [27].

Impellizzeri et al.’s study suggests that using DM with 0.014 × 0.025 CuNiTi wires in a self-ligating straight-wire appliance successfully reduced the number of appointments for each patient from 3 appointments in 10 weeks to 2 per 10 weeks. Naturally, a reduction in chair time and material costs was observed. Moreover, more precise evaluation of treatment by the doctor was possible [48].

Another study by Sangalli et al. revealed that when patients were equipped with a cheek retractor and scan box by Dental Monitoring and instructed to take monthly intra-oral scans, this study group of patients showed a significant improvement in plaque control compared to the control group. A decreased number of emergency appointments in the study group was also registered, although it was not significant. The patients were not orthodontic treatment cases [52].

Maspero et al., in a 2020 article, confirmed that this application saved 5.8 appointments over a 2-year treatment. Its software platform was observed by patients as user-friendly and they noted improvement of communication with the doctor. Moreover, it was observed that stability of the result could also be measured and, if relapse of misalignment of the teeth were to occur, swift measures could be enacted to interfere with relapse development. Measurement was carried out by Moylan et al. in 2019. They compared intercanine and intermolar measurement differences between plaster models based on impressions taken by a dentist versus measurements from data from Dental Monitoring software. The differences ranged from 0.17 mm and −0.02 mm; this was assessed as sufficiently accurate [34,54]. Another publication measured the difference in STL (Stereolithography) files provided from the iTero scanners and STL files generated by DM software. Differences ranged from 0.0148 mm to 0.0275 mm; thus, they safely stated the method is accurate enough for clinical applications [55].

### 3.3. Other AI and Teldentistry Applications

An article by Achmad utilized teledentistry in order to connect with distant patients for consultations throughout the COVID-19 pandemic, which was exceptionally well-accepted by both groups [53]. Another purpose of teledentistry was documented by Deshpande et al., who found out that if trained general dentists were remotely communicating with orthodontists via teledentistry, more accurate interceptive orthodontic treatments would be made available, which thus led to a reduction in severity of malocclusions in disadvantaged children where referral was not plausible. Moreover, unnecessary referrals were filtered out, which is an advantage both for specialist and patient. They also warn of the risk that diagnosis based on clinical photography made by the patient may not be accurate, and the practitioner may not collect all necessary data, since other diagnostic procedures such as percussion and palpation cannot be performed via photography [41].

Asiri et al. summarized in their review that most commonly utilized AI domains were intended for diagnostics and treatment planning, followed by automated anatomic landmark detection used for cephalometry. Marginally, AI was used for assessment of growth and development and evaluation of treatment outcome [39].

## 4. Discussion

Contemporary data show that there is a growing trend towards the use of telemedicine in modern dental and orthodontic practices, as it has been proven to increase efficiency and allow dentists to specifically monitor each case and focus on the most important goals for each patient, while saving chairside time and patient resources and preventing deterioration of their dental status, from which they also benefit financially, psychologically, and esthetically. Likewise, the quality of treatment is improved, and the time needed to resolve problems is reduced. Although the benefits for the patient are not yet fully known, since the willingness to use the modern aids is not yet as high as the doctor would like, the demand is increasing [11,25,25,30,32,42,51,56,57].

The availability for the patient and the practice is indeed very high: the only technical requirement for the patient is a smartphone and an internet connection. The rest is provided by the dental practice. Patient compliance is also statistically higher. Undoubtedly, more and more applications will be developed to facilitate the treatments even more and increase the comfort for the user during the treatment [43,45,54,58,59].

In comparison with conventional methods of treatment management, physicians will be able to increase the number of patients they can treat at one time without compromising the quality of the services provided. They can, in fact, observe the patient’s dental status more often, with great detail, instruct the patient remotely to aid in his or her treatment, or change instructions for further steps, e.g., change of placement of intra- and inter-maxillary elastics. They do not have to rely on a patient´s observation skills in terms of debonding of brackets, attachments, or other auxilliars, and problems can be detected much sooner than 3–4 weeks of the next appointment. Moreover, the treatment doctor can very quickly detect the first signs of relapse of the malocclusion, even at the slightest movement of a singular tooth. In addition, improved compliance is observed through the use of the new, attractive AI software. Finally, the ability to seek treatment over long distances is highly desirable, both for patients who travel frequently or live abroad and during pandemics such as those that have occurred in recent years [12,14]. Additionally, when comparing traditional diagnostic methods, the use of AI systems can speed up and enhance even the development of complicated orthognathic surgery treatment planning, where fast cephalometric tracing is performed by software. Jaw segmentation is also faster when performed by AI then by even skilled practitioners. Furthermore, dental and skeletal pathologies can be detected easier and not be omitted from an orthodontist’s line of sight, as during cephalometric analysis his focus is mostly on the landmarks and bigger picture of a patient´s skeleton rather than singular teeth.

The combination of AI and teledentistry introduces a historical paradigm shift in orthodontic care. Software enhanced by advanced AI provides not only sophisticated evaluations of clinical situation and post-treatment stability but also pre-treatment diagnostics or even automated segmentations of CBCT utilized for cephalometric [36,60,61,62,63,64,65,66,67], airway [68,69], or forensic applications [70,71]. AI-powered software for orthodontic cephalometric analysis recently became a common tool for a reliable and accurate cephalometric tracing method [61,72], which represents a significant evolution from the times of analog cephalometric processing [73].

This review identified several limitations to using AI-powered systems in orthodontics:Accuracy: AI-powered systems can help with diagnosis and treatment planning, but they are not as accurate as a trained orthodontist in identifying and treating complex cases [55,74,75], although some reports have shown that the level of accuracy is nearing the human level.Expertise: AI systems do not have the same level of clinical expertise as a trained orthodontist. They may not be able to fully understand the patient’s needs and cannot provide the same level of individualized care [30,48,74].Ethical concerns: There are also ethical concerns about the use of AI in healthcare, including the possibility of biased algorithms and the potential to replace human labor with automation [76,77,78].Cost: AI systems can be expensive to implement and maintain and may not be accessible to all patients or clinicians.Regulation: the use of AI in healthcare also comes with regulatory challenges. These include the need for oversight to ensure the accuracy and safety of AI-powered systems [11,79,80].

A limitation of this paper is that there is a wide range of different attributes and parameters that could be used to evaluate the benefits to both parties, and further studies should be conducted that explore each parameter in more depth.

This paper also highlights that the use of AI software in orthodontics raises questions about reliability, as these tools can contain errors and bias that can lead to mistakes or mishaps during treatment. The review included most impactful studies on the use of AI in orthodontics and summarized the characteristics of current software alternatives. The accuracy and expertise have been evaluated as sufficient, albeit a sufficient number of studies on this matter have not been published yet. The value of AI-powered monitoring of the orthodontic retention phase is not completely appreciated yet and very few studies are focusing on this aspect. The authors of this paper see unexplored potential in this direction.

Current clinical decision support systems in orthodontics are already supported by AI. Commercial companies that manufacture clear aligners use data from millions of digital intraoral scans sent by clinical providers and apply AI algorithms to predict and plan tooth movement after they perform tooth segmentation. However, such AI algorithms are not validated and require clinicians to exercise caution when using the predictions provided and monitoring treatment outcomes [81,82].

Scientific reviews mapping the clinical application of orthodontic AI show a significant increase since 2020, recognizing the potential to support the assessment of orthodontic treatment and retention in a variety of ways. This has been accelerated by the global pandemic and technological AI breakthroughs. In 2022, AI algorithms were used to analyze and interpret digital images and diagnostic data, such as dental radiographs, photographs, CBCT, or intraoral photos and video scans, to identify problems and predict treatment course, outcome, or stability. AI has also been widely used to monitor patients during treatment and provide real-time feedback and alerts to ensure treatment is going as planned. AI-based systems and their application have even reached university orthodontic curricula [12,14,26,68,83,84,85,86,87,88,89,90,91,92].

In addition, AI can be used to help orthodontists track and analyze patient data over time, allowing them to identify trends and patterns that may be useful in predicting treatment outcomes and optimizing treatment plans. This could be especially useful for patients with complex or difficult cases, where traditional methods of assessment may not be sufficient [25,27,27,32].

Non lege artis treatment can take many forms, such as using treatments that have not been proven effective, using treatments in an inappropriate or dangerous manner, or failing to follow accepted protocols for diagnosing or treating a particular condition. Such treatment may also involve exploitation or abuse of patients, such as taking advantage of their vulnerability or trust. AI implementations in orthodontic software are no exception.

In 2023, the European Union announces the idea of creating the world’s first comprehensive standards for regulating or prohibiting certain applications of artificial intelligence [79].

The EU’s AI law is expected to lead the world in regulating AI. The AI Act is a proposal for a European law on artificial intelligence (AI)—the first law on AI to be passed by a major regulator. The law assigns applications of AI to three categories of risk. First, applications and systems that pose unacceptable risk, including state-run social assessments such as those used in China, are banned. Second, high-risk applications, such as a CV scanning tool that ranks job applicants, are subject to specific legal requirements. Finally, applications that are not explicitly banned or classified as high-risk are largely unregulated [93].

## 5. Conclusions

The use of AI in the assessment and retention of orthodontic treatment is an emerging area with significant potential for improving patient care and outcomes. It is likely to see many more AI-powered tools and systems being developed and adopted in the field of orthodontics in the coming years.

Literature research concludes that while AI-powered systems already effectively assist in orthodontic treatment, they must be used in conjunction with properly trained orthodontists to achieve the best possible outcomes for patients. Unsupervised applications of AI-assisted systems in orthodontics are not in accordance with the standards of good medical practice or the principles of medical ethics. With current unresolved risks of AI bias and incoming AI governmental regulations, such an unsupervised orthodontic treatment would be considered as non lege artis.

This scoping review proves that the current clinical adoption of AI-powered systems has already reshaped the form of modern orthodontic practice, albeit they are still rife with limitations such as: accuracy, expertise, ethical concerns, cost, and regulatory issues.

## Figures and Tables

**Table 1 healthcare-11-00683-t001:** The most cited articles relevant to the queried keywords in researched topic.

#	Authors	Title	Citations	FWCI	Reference	Published
1	Kunz et al.	Artificial intelligence in orthodontics: Evaluation of a fully automated cephalometric analysis using a customized convolutional neural network	65	12.89	[33]	2020
2	Maspero et al.	Available technologies, applications and benefits of teleorthodontics. A literature review and possible applications during the COVID-19 pandemic	59	3.44	[34]	2020
3	Yu et al.	Automated Skeletal Classification with Lateral Cephalometry Based on Artificial Intelligence	57	10.21	[35]	2020
4	Lee et al.	Automated cephalometric landmark detection with confidence regions using Bayesian convolutional neural networks	40	7.51	[36]	2020
5	Leite et al.	Radiomics and Machine Learning in Oral Healthcare	38	1.83	[37]	2020
6	Wang et al.	Multiclass CBCT Image Segmentation for Orthodontics with Deep Learning	27	11.36	[38]	2021
7	Bichu et al.	Applications of artificial intelligence and machine learning in orthodontics: a scoping review	20	6.35	[39]	2021
9	Schwendicke et al.	Deep learning for cephalometric landmark detection: systematic review and meta-analysis	18	3.06	[40]	2021
10	Deshpande et al.	Teledentistry: A boon amidst COVID-19 Lockdown—A narrative review	16	1.67	[41]	2021
11	Ahmed et al.	Artificial Intelligence Techniques: Analysis, Application, and Outcome in Dentistry—A Systematic Review	16	1.34	[42]	2021
12	Mohammadad-Rahimi et al.	Machine learning and orthodontics, current trends and the future opportunities: A scoping review	14	4.34	[43]	2021
13	Juerchott et al.	In vivo comparison of MRI- and CBCT-based 3D cephalometric analysis: beginnning of a non-ionizing diagnostic era in craniomaxillofacial imaging?	14	1.44	[44]	2020
14	MacHoy et al.	The ways of using machine learning in dentistry	14	0.84	[45]	2020
15	Ren et al.	Machine learning in dental, oral and craniofacial imaging: A review of recent progress	13	1.39	[46]	2021
18	Dalessandri et al.	Attitude towards telemonitoring in orthodontists and orthodontic patients	11	4.8	[47]	2021
20	Caruso et al.	A knowledge-based algorithm for automatic monitoring of orthodontic patients: The dental monitoring system. Two cases	10	1.81	[27]	2021
21	Impellizzeri	Dental Monitoring Application: I tis a valid innovation in the Orthodontics Pracice?	9	0.86	[48]	2020
22	Monill-González et al.	Artificial intelligence in orthodontics: Where are we now? A scoping review	9	2.58	[49]	2021
23	Thurzo et al.	Where Is the Artificial Intelligence Applied in Dentistry? Systematic Review and Literature Analysis	8	5.15	[12]	2022
24	Bulatova et al.	Assessment of automatic cephalometric landmark identification using artificial intelligence		3.72	[50]	2021
25	Park et al.	Teledentistry platforms for orthodontics	8	3.39	[51]	2021
26	Sangalli et al.	Effects of remote digital monitoring on oral hygiene of orthodontic patients: a prospective study	7	3.05	[52]	2021
27	Achmad et al.	Teledentistry as a solution in dentistry during the covid-19 pandemic period: A systematic review	6	0.78	[53]	2020

## Data Availability

Not applicable.

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
