# Peer review of "Artificial Intelligence Systems Assisting in the Assessment of the Course and Retention of Orthodontic Treatment"

_healthcare, 2023, doi:10.3390/healthcare11050683_

Round 1
Reviewer 1 Report
The Authors conducted a scoping review aiming at exploring the application of advanced AI software to be used in order to improve clinical protocols in orthodontics. I have some suggestions that the Authors might want take into account in order to improve the quality of their manuscript.
1. English usage and style need to be fixed.
2. Include in the Abstract section the methodology of your research.
3. Consider cropping the introduction section as it appears too long and a little confusing.
4. Consider including among the searched databases also Cochrane Library.
5. What is the rationale for not using the PRISMA protocol to conduct your review? Tricco, AC, Lillie, E, Zarin, W, O'Brien, KK, Colquhoun, H, Levac, D, Moher, D, Peters, MD, Horsley, T, Weeks, L, Hempel, S et al. PRISMA extension for scoping reviews (PRISMA-ScR): checklist and explanation. Ann Intern Med. 2018,169(7):467-473. doi:10.7326/M18-0850.; Moher D, Shamseer L, Clarke M, Ghersi D, Liberati A, Petticrew M, Shekelle P, Stewart LA. Preferred Reporting Items for Systematic Review and Meta-Analysis Protocols (PRISMA-P) 2015 statement. Syst Rev. 2015;4(1):1. doi: 10.1186/2046-4053-4-1; Shamseer L, Moher D, Clarke M, Ghersi D, Liberati A, Petticrew M, Shekelle P, Stewart LA, the PRISMA-P Group. Preferred Reporting Items for Systematic Review and Meta-Analysis Protocols (PRISMA-P) 2015: elaboration and explanation. BMJ 2015.349:g7647. doi: 10.1136/bmj.g7647
6. Please, describe the PICOS question in a more extended way in the text in order to give a more accurate question of research to the reader.
7. The Result section is completely missing. In the section results it is highly recommended to report raw numbers of the examined studies from the first research. In addition, it is important to describe how many articles were excluded in each step and why.
8. In the discussion section, a clear statement of the main limitations of the study cannot be found.
Overall, the manuscript needs extensive revision before being suitable for publication.
Author Response
Reviewer 1
Comments and Suggestions for Authors
Extensive editing of English language and style required.
The Authors conducted a scoping review aiming at exploring the application of advanced AI software to be used in order to improve clinical protocols in orthodontics. I have some suggestions that the Authors might want take into account in order to improve the quality of their manuscript.
- English usage and style need to be fixed.
2. Include in the Abstract section the methodology of your research.
3. Consider cropping the introduction section as it appears too long and a little confusing.
4. Consider including among the searched databases also Cochrane Library.
5. What is the rationale for not using the PRISMA protocol to conduct your review? Tricco, AC, Lillie, E, Zarin, W, O'Brien, KK, Colquhoun, H, Levac, D, Moher, D, Peters, MD, Horsley, T, Weeks, L, Hempel, S et al. PRISMA extension for scoping reviews (PRISMA-ScR): checklist and explanation. Ann Intern Med. 2018,169(7):467-473. doi:10.7326/M18-0850.; Moher D, Shamseer L, Clarke M, Ghersi D, Liberati A, Petticrew M, Shekelle P, Stewart LA. Preferred Reporting Items for Systematic Review and Meta-Analysis Protocols (PRISMA-P) 2015 statement. Syst Rev. 2015;4(1):1. doi: 10.1186/2046-4053-4-1; Shamseer L, Moher D, Clarke M, Ghersi D, Liberati A, Petticrew M, Shekelle P, Stewart LA, the PRISMA-P Group. Preferred Reporting Items for Systematic Review and Meta-Analysis Protocols (PRISMA-P) 2015: elaboration and explanation. BMJ 2015.349:g7647. doi: 10.1136/bmj.g7647
6. Please, describe the PICOS question in a more extended way in the text in order to give a more accurate question of research to the reader.
7. The Result section is completely missing. In the section results it is highly recommended to report raw numbers of the examined studies from the first research. In addition, it is important to describe how many articles were excluded in each step and why.
8. In the discussion section, a clear statement of the main limitations of the study cannot be found.
Overall, the manuscript needs extensive revision before being suitable for publication.
Dear Reviewer,
Thank you for your comments and suggestions. We have thoroughly revised the English language and style in this article. As you suggest in your first comment. We had included the research methodology in the abstract and have rewritten it to make the text more fluid. We have shortened the introduction and rewritten various parts to make it clearer.
Regarding your comment that you should consider "including among the searched databases also Cochrane Library" our point is that while the Cochrane Library is a highly regarded source of systematic reviews and meta-analyses of clinical interventions, it is not the most appropriate source for a scoping review. Our scoping review is neither a clinical intervention nor does it contain a meta-analysis. And like any scoping review, we have a broader framework that aims to identify all relevant literature on a topic rather than summarize the existing evidence. There are several reasons why the Cochrane Library is not the best choice for a scoping review, and a generative AI like chat chatGPT can explain this in more detail. To keep our explanation brief, we believe that while the Cochrane Library is an excellent source for systematic reviews of clinical interventions, it is not the best choice for a scoping review.
Regarding your question about the reasons for not using the PRISMA protocol in conducting your review': we had previously published the systematic review (https://www.mdpi.com/2227-9032/10/7/1269) where we used PRISMA protocols. PRISMA is a well-known and widely used protocol for conducting systematic reviews of primary research studies. However, a review has a different focus and purpose than a systematic review, and therefore PRISMA is not the most appropriate protocol for a review.
Dear reviewer, in relation to your question about the description of PICOS - scoping reviews do not usually use the PICOS (Population, Intervention, Comparison, Outcome, Study design) framework to develop a specific research question, as is common with systematic reviews. Instead, scoping reviews often begin with a broad research question, such as "What is known about a particular topic?" or "What is the scope and nature of research on a particular topic?" The aim of a scoping review is to survey the existing literature and provide an overview of the available evidence, not to answer a specific research question as in a systematic review.
Regarding your comment about the missing Results section. Following your recommendation, we have listed the results in a separate table and created the Results section. The purpose of a scoping review is to gather the existing literature on a specific topic. The Results section now provides an overview of the studies that were included in the review, the characteristics of these studies and the main themes or findings that emerged from the review are explained in Discussion.
There is no clear statement of the main limitations of the study in the Discussion section. Limitations were stated in lines 413-435, with a focus on the limitations of this study in lines 432-435.
Reviewer 2 Report
The authors have dealt with a topic subject to innovation in the dental world, the work, despite listing some aspects, does not incisively highlight the real objective.
- list which verifiable parameters can be evaluated by a software.
- better specify what the real advantage is by comparing it with traditional methods (data collection) not forgetting the medico-legal parameter
Author Response
Reviewer 2
The authors have dealt with a topic subject to innovation in the dental world, the work, despite listing some aspects, does not incisively highlight the real objective.
- list which verifiable parameters can be evaluated by a software.
- better specify what the real advantage is by comparing it with traditional methods (data collection) not forgetting the medico-legal parameter
Dear reviewer,
Thank you very much for your constructive feedback. We respected your point of view and added and reworked the discussion section which now much more clearly lists the parameters and possibilities that can be evaluated, and it is also compared and to the conventional methods and explaining and listing the advantages of AI powered orthodontics workflow.
We also added a whole result section so the whole review is more comprehensible.
Reviewer 3 Report
TITLE: Artificial Intelligence Systems Assisting in the Assessment of 2 the Course and Retention of Orthodontic Treatment
scoping review
The purpose of the authors is to explore applications of advanced AI software that can potentially improve today's daily working protocols in orthodontics as well as their limitations. Emphasis is on evaluation of the accuracy and efficiency of current AI-powered systems compared to conventional methods for assessment of patient's treatment progress and its stability in the follow-up period.
In the review the authors have identified a trend in a rapid development of various software aimed at meeting the demand for a quick-to-use, accurate, and practical tool for orthodontists that could improve their workflow and other aspects of the care in this field.
Thy stated that: (a) such an instrument allows the practitioner to detect the stability of completed treatments, changes in occlusion, jaw displacements and, finally, almost imperceptible movements of all teeth, which can ultimately be crucial for the micro- and macro- esthetics of the patient's smile and face. (b)However, the question arises as to whether this data is accurate enough to be trusted by the practice.
The study is interesting.
However, it needs some improvements before being considered for acceptance.
Among the strengths we find:
-The authors deal with an innovative and interesting theme.
-The text has been developed in a very understandable way even for those who have no experience in data science in the development of artificial intelligence algorithms.
Among the weaknesses we find:
-The authors should make the achievements of their review emerge better, accompanying them with a brief analysis of what emerges in the works taken into consideration.
-The limitations that emerge (see page 5) seem to be suddenly inserted in the text. They should be accompanied by a description of what emerges in the cited works.
- It would be necessary to use editorial tools such as tables to better outline what emerges.
Others:
1. The abstract must summarize better the sections. It directly starts with the purpose.
2. The purpose is very badly written. See “The purpose of this paper was to review the utilizations of advanced AI software in orthodontics, particularly for the purposes of assessment of treatment progress and ou come stability in the follow-up phase. To ….The secondary..” Insert the key questions the review must answer. Use bullet points.
3. Some sentences must be smoothed. See for example “This review identified a trend in a rapid development of various software aimed at meeting the demand for a quick-to-use, accurate, and practical tool for orthodontists that could improve their workflow by enabling them to thoroughly monitor each patient, set a specific desired outcome for that patient in a pre-estimated time frame, track its fulfillment, notify the practitioner of a delay in achieving the set goals, warn of a possible worsening of pre-existing pathologies such as gum recession, gingivitis, enamel defects, etc.”
4. I suggest to insert the section results with the true outocome of the review.
5. The discussion must be dedicated to the comparison to other similar studies, to the limitation of the investigated studies, and to the limitation of their review.
6. Use tables to list the works with a brief summary of one of two sentences.
7. The paragraph “In 2023, the European Union announces to create the world's first comprehensive standards for regulating or prohibiting certain applications of artificial intelligence. EU AI law 229 is expected to lead the world in regulating AI [64].” has two problems. The first problem is that the first sentence is not supported by a reference. The second problem is that the second sentence is both
Author Response
Reviewer 3
TITLE: Artificial Intelligence Systems Assisting in the Assessment of 2 the Course and Retention of Orthodontic Treatment
scoping review
The purpose of the authors is to explore applications of advanced AI software that can potentially improve today's daily working protocols in orthodontics as well as their limitations. Emphasis is on evaluation of the accuracy and efficiency of current AI-powered systems compared to conventional methods for assessment of patient's treatment progress and its stability in the follow-up period.
In the review the authors have identified a trend in a rapid development of various software aimed at meeting the demand for a quick-to-use, accurate, and practical tool for orthodontists that could improve their workflow and other aspects of the care in this field.
Thy stated that: (a) such an instrument allows the practitioner to detect the stability of completed treatments, changes in occlusion, jaw displacements and, finally, almost imperceptible movements of all teeth, which can ultimately be crucial for the micro- and macro- esthetics of the patient's smile and face. (b)However, the question arises as to whether this data is accurate enough to be trusted by the practice.
The study is interesting.
However, it needs some improvements before being considered for acceptance.
Among the strengths we find:
-The authors deal with an innovative and interesting theme.
-The text has been developed in a very understandable way even for those who have no experience in data science in the development of artificial intelligence algorithms.
Among the weaknesses we find:
-The authors should make the achievements of their review emerge better, accompanying them with a brief analysis of what emerges in the works taken into consideration.
-The limitations that emerge (see page 5) seem to be suddenly inserted in the text. They should be accompanied by a description of what emerges in the cited works.
- It would be necessary to use editorial tools such as tables to better outline what emerges.
Others:
- The abstract must summarize better the sections. It directly starts with the purpose.
- The purpose is very badly written. See “The purpose of this paper was to review the utilizations of advanced AI software in orthodontics, particularly for the purposes of assessment of treatment progress and ou come stability in the follow-up phase. To ….The secondary..” Insert the key questions the review must answer. Use bullet points.
- Some sentences must be smoothed. See for example “This review identified a trend in a rapid development of various software aimed at meeting the demand for a quick-to-use, accurate, and practical tool for orthodontists that could improve their workflow by enabling them to thoroughly monitor each patient, set a specific desired outcome for that patient in a pre-estimated time frame, track its fulfillment, notify the practitioner of a delay in achieving the set goals, warn of a possible worsening of pre-existing pathologies such as gum recession, gingivitis, enamel defects, etc.”
- I suggest to insert the section results with the true outocome of the review.
- The discussion must be dedicated to the comparison to other similar studies, to the limitation of the investigated studies, and to the limitation of their review.
- Use tables to list the works with a brief summary of one of two sentences.
- The paragraph “In 2023, the European Union announces to create the world's first comprehensive standards for regulating or prohibiting certain applications of artificial intelligence. EU AI law 229 is expected to lead the world in regulating AI [64].” has two problems. The first problem is that the first sentence is not supported by a reference. The second problem is that the second sentence is both
Dear Reviewer,
We are pleased that you also find our paper interesting. We also thank you for your kind comments on the strengths of our paper and your pragmatic suggestions. In the following, we address all your concerns about the weaknesses of our study and describe the changes we have made based on your recommendations.
We have rewritten parts of Introduction and Discussion and created a separate Result chapter with a new table to better outline what emerges from the results with a concise analysis of what arises in the papers taken into consideration. We have added a brief description of what develops in the cited works.
- The abstract must summarize better the sections. It directly starts with the purpose.
We have rewritten the whole abstract as you have suggested.
- The purpose is very badly written. See “The purpose of this paper was to review the utilizations of advanced AI software in orthodontics, particularly for the purposes of assessment of treatment progress and ou come stability in the follow-up phase. To ….The secondary..” Insert the key questions the review must answer. Use bullet points.
We have rewritten the whole paragraph, albeit we have avoided using the bulleted list intentionally. Lines 230-237
- Some sentences must be smoothed. See for example “This review identified a trend in a rapid development of various software aimed at meeting the demand for a quick-to-use, accurate, and practical tool for orthodontists that could improve their workflow by enabling them to thoroughly monitor each patient, set a specific desired outcome for that patient in a pre-estimated time frame, track its fulfillment, notify the practitioner of a delay in achieving the set goals, warn of a possible worsening of pre-existing pathologies such as gum recession, gingivitis, enamel defects, etc.”
Thank you, we have rewritten the whole paragraph.
- I suggest to insert the section results with the true outocome of the review.
Thank you for your suggestion, we have inserted section Results with a separate table. rewritten the whole paragraph.
- The discussion must be dedicated to the comparison to other similar studies, to the limitation of the investigated studies, and to the limitation of their review.
We have narrowed the Discussion with focus on similar studies and limitations.
- Use tables to list the works with a brief summary of one of two sentences.
We have created the table, albeit we have avoided creating Tldr;s for them intentionally.
- The paragraph “In 2023, the European Union announces to create the world's first comprehensive standards for regulating or prohibiting certain applications of artificial intelligence. EU AI law 229 is expected to lead the world in regulating AI [64].” has two problems. The first problem is that the first sentence is not supported by a reference. The second problem is that the second sentence is both
Thank you, we have reorganized the paragraph, elaborated current status and added one new online reference (102) .
Round 2
Reviewer 1 Report
Dear Authors,
thank you for having addressed all the requests raised during my first review round. I am satisfied with the given answers and the improvement made to your manuscript.
Kind regards
Reviewer 3 Report
I thank the authors for including all the suggestions and for having answered to the reliefs in a complete way.
I have no comments.